# Recent Advances in Plasma-Based Cancer Treatments: Approaching Clinical Translation through an Intracellular View

Elahe Alizadeh [1,*] and Sylwia Ptasińska [2,3]

1 Department of Medicine, Queen's University, Kingston, ON K7L 3J9, Canada
2 Radiation Laboratory, University of Notre Dame, Notre Dame, IN 46556, USA; Sylwia.ptasinska.1@nd.edu
3 Department of Physics, University of Notre Dame, Notre Dame, IN 46556, USA
* Correspondence: Elahe.Alizadeh@queensu.ca

**Abstract:** Plasma medicine is a multidisciplinary field of research which is combining plasma physics and chemistry with biology and clinical medicine to launch a new cancer treatment modality. It mainly relies on utilizing low temperature plasmas in atmospheric pressure to generate and instill a cocktail of reactive species to selectively target malignant cells for inhibition the cell proliferation and tumor progression. Following a summarized review of primary in vitro and in vivo studies on the antitumor effects of low temperature plasmas, this article briefly outlines the plasma sources which have been developed for cancer therapeutic purposes. Intracellular mechanisms of action and significant pathways behind the anticancer effects of plasma and selectivity toward cancer cells are comprehensively discussed. A thorough understanding of involved mechanisms helps investigators to explicate many disputes including optimal plasma parameters to control the reactive species combination and concentration, transferring plasma to the tumors located in deep, and determining the optimal dose of plasma for specific outcomes in clinical translation. As a novel strategy for cancer therapy in clinical trials, designing low temperature plasma sources which meet the technical requirements of medical devices still needs to improve in efficacy and safety.

**Keywords:** low temperature plasma; cancer treatment; radiation therapy; in vitro and in vivo studies; reactive species; apoptosis; mitochondria; oxidative damage; plasma oncology

## 1. Background and Motivation

Cancer is the second leading cause of morbidity and mortality worldwide after cardiovascular diseases [1]. Although, significant progresses have been achieved toward a better understanding of cancer therapy over the last few decades, nevertheless the World Health Organization estimated that this family of diseases was responsible for a likely 9.6 million deaths in 2018, which was about 1 in 6 deaths in the world [2]. Thus, cancer is still considered one of the deadliest threats to human health. Current conventional cancer treatments, comprising of surgery, chemotherapy and radiation therapy, all aim to achieve a complete eradication of cancer cells without affecting non-malignant tissues. In surgery, complete surgical excision of tumor cells is challenged by microscopic tumor residues, consequently tumors can return if not fully removed. In radiation therapy (exploiting high-energy ionizing radiation), the necessity to protect healthy tissues surrounding a tumor is the major issue that extremely limits the therapeutic radiation dose. Despite of causing inevitably damages to normal tissues, radiation therapy still remains as an important modality for curing at least 50% of all cancer patients [3]. Similarly, in chemotherapy (utilizing cytotoxic drugs), chemotherapeutic agents point cells with the high basal level of proliferation and regeneration. Thus, both tumor cells and surrounding healthy cells with rapid proliferation (like hair, skin, bone marrow and epithelium of the gastrointestinal tract cells) are targeted by agents, consequently causing highly toxic effects related to the treatment [4,5]. In combined-modality therapy, cancer patients are treated with two or more of these modalities rather than just one to improve the chance of cure. For instance,

chemoradiation therapy (CRT) is combining of radiation therapy and chemotherapy at the same time. Moreover, in concurrent chemoradiation therapy (CCRT), employing radiosensitizer or radioenhancers with platinum chemotherapeutic drugs together sensitizes the malignant cells to high-energy radiation [6] which is offering an alternative application of chemoradiation therapy [7,8]. Some emerging strategies including photothermal therapy (using photothermal agents activated by light to produce heat for tumor destruction), gene therapy and immune therapy have also shown reasonable potential in cancer treatment. However, they undergo various limitations like drug resistance, pathogenesis complications, cytotoxicity to healthy tissues, inadequate delivery methods to the tumor site, and high recurrence rates of some certain types of cancer. In general, the ideal alternatives and more effective therapies for cancer should be less-invasive treatments with strong cytotoxic effect on malignant cells and inferior side effects on healthy cells.

One of the most crucial causes of cancer initiation and progression is the formation of reactive species in biological systems. Interestingly, on the other hand, high doses of reactive species possess the capability to damage malignant cells. Therefore, utilizing external physical or chemical methods to produce and instill a high concentration of reactive species to the cells is a promising approach for inhibition the growth of cancer cells via several intracellular mechanisms. Low temperature plasma (LTP) is another new modality for cancer treatment relying on the generating a cocktail of reactive species in plasma to selectively target malignant cells for inhibition and treatment of cancer. Understanding of the anticancer mechanisms of plasma-based processes and LTP's selectivity toward cancer cells still need to be investigated [9].

Plasma, called the forth state of the matter, fundamentally is an entirely or partially ionized gas consisting of biologically and chemically reactive species, including free electrons and radicals, as well as atoms, and molecules either in neutral or charged form. In addition to chemically reactive species, depending on plasma force, plasma produces physically active agents, i.e., electromagnetic fields leading to the emission of visible light, ultraviolet (UV) or vacuum ultraviolet (VUV) radiations, and propagating disturbances like shock waves and heating [10]. In the laboratory, plasmas are conveniently created by applying an electric field to the injected gas or vapor between two electrodes, typically pure helium, argon, neon or their mixtures with different percentages of oxygen or other compounds. The electric field accelerates electrons and initiates a cascade of collisional processes (excitation, ionization, and dissociation) that gives rise to a diverse range of chemical species. The numbers of positively charged ions and electrons in the discharge are generally equal except in plasma surfaces, where electric fields are strong. The amount of applied power, and the type and pressure of the processing gas determine the energy (thus the temperature) and the chemical combination of the cocktail of species. Proportionality of electrons and positive ions results in no relatively high electric charge at low pressures (like in fluorescent lamps) or at very high temperatures (like in stars and nuclear fusion reactors). At near-ambient temperatures or in low temperature plasma (so-called cold plasma, non-thermal plasma or non-equilibrium plasma), the gas temperature is slightly above room temperature and biologically tolerable ($< 40\ ^\circ$C), while electron temperature is in order of a few thousands of $^\circ$C [11,12]. Low temperature plasmas applied in atmospheric pressure are efficient sources of very high concentrations of reactive species. They contain reactive atomic and molecular species, including unstable, short lived or metastable excited atoms or ions and radicals. Moreover, they prominently contain reactive oxygen species (ROS) such as hydroxyl free radicals (OH) and hydrogen peroxide ($H_2O_2$), and reactive nitrogen species (RNS) such as nitric oxide (NO) and nitrite ($NO_2^-$) [13,14].

The complex nature and exact mechanisms of interaction of atmospheric pressure LTP with biological systems has been intensively investigated [11,15–19]. Most of these studies have focused on the role of reactive species in cancer initiation and progression, as well as their antitumor effects in a variety of malignance, consequently indicating the capacity of LTP to induce highly significant lethal effects in cells and cancer treatment [20]. In 2004, based on some primary results, Stoffels et al. [21] introduced the basis for a novel

interdisciplinary field of research later called plasma medicine: optimal applications of LTP and its therapeutic potential in medicine and biology [22,23]. Friedman and Keidar were among the pioneering researchers who developed LTP sources for medical applications and showed that cold plasma selectively kills cancer cells. Friedman et al. used cold plasma for cancer treatment and showed that high doses of plasma leads to necrosis death and low doses to initiate apoptotic death post treatment [24–27]. Over the next decade after these findings, numerous in vitro studies have performed and showed remarkable selective anticancer effects of non-thermal atmospheric pressure plasma on approximately 20 types of malignant cell lines; including lung cancer, hepatocellular carcinoma, glioblastoma, prostate cancer, ovarian cancer, osteosarcoma, melanoma and breast cancer [28]. Table 1 summarizes the most milestones in in vitro studies in the development of plasma-based methods in a chronological order. Various type of LTP devices were used to directly irradiate different cancer cells cultured in the multi-well plates or Petri dishes. As abridged as key results in the last column of Table 1, in vitro studies mainly exhibit antitumor selectivity and anti-metastatic activity of LTP, proposing wide applicability of plasma in the treatment of various cancer types as melanoma, colon cancer, lung cancer, hepatocellular carcinoma, pancreatic, prostate and breast cancer (See Table 1 and references therein). It is determined that employing plasma on cellular systems is producing a plethora of reactive species; their interactions with cells can manipulate cells' redox signaling and ultimately cause a variety of cellular responses including the alteration of surface receptor functions, induction of cell cycle arrest, and DNA damage-induced activation of p53 followed by a subsequent p53-dependent apoptosis [10,29,30].

**Table 1.** Summary of in vitro studies in chronological order, exploiting different types of plasma sources (devices) on various cell lines (study models) relevant to cancer treatment, along with the key results obtained from each study.

| Year of Study | Study Group [Refs] | Plasma Device Type (Injected Gas) | Study Model (Cancer Cell Type) | Key Results |
|---|---|---|---|---|
| 2006–2007 | **Fridman et al. [24,25]** | FE-DBD plasma (Air) | Human melanoma skin cancer cells (A2058) | Promoting apoptotic behavior in cancer cells |
| 2008 | **Zhang et al. [31]** | Plasma jet (Ar) | Human hepatocellular liver cancer cells (BEL-7402) | Enhancing the apoptosis activity in cancer cells after adding oxygen to plasma |
| 2009 | **Lee et al. [32]** **Kim et al. [33]** | Radio-frequency plasma jet (He) | Human melanoma skin cancer cells (G361) | Inhibition of the malignant transformation and halt on cancer metastasis, death of melanoma cells |
| 2010 | **Lupu et al. [34]** **Georgescu et al. [35]** | Plasma jet (He) | Human colorectal cancer cells (COLO 320DM) Murine melanoma cells (B16) | Higher apoptotic behavior in cancer cells |
| 2010 | **Kim et al. [36]** | Spray torch NTP jet (He) | Human colorectal cancer cells (HCT116, SW480, LoVo) | Anti-proliferative activity and halt on cancer metastasis |
| 2010 | **Zirnheld et al. [37]** | NTP jet (He) | Human melanoma cells (1205Lu) | Significant death of melanoma cells |
| 2011 | **Ahn et al. [38]** | Micro-nozzle array plasma jet (Air and $N_2$) | Human cervical carcinoma cells (HeLa) | Dysfunction of mitochondria and initiating mitochondria-mediated apoptosis |
| 2011 | **Kalghatgi et al. [39]** **Sensenig et al. [40]** | NTP-DBD (Air) | Mammalian breast epithelial cells (MCF10A) Human melanoma cancer cells Murine melanoma cells (B16-F10) | Replication arrest or formation of single-stranded DNA breaks and induction of apoptosis |
| 2011 | **Keidar et al. [27]** | Plasma jet (He) | Normal human bronchial epithelial (NHBE) Lung cancer cell (SW900) | Higher apoptosis and decreasing cell migration velocity and metastasis |

**Table 1.** *Cont.*

| Year of Study | Study Group [Refs] | Plasma Device Type (Injected Gas) | Study Model (Cancer Cell Type) | Key Results |
|---|---|---|---|---|
| 2011 | **Kim et al. [41]** | Micro-size plasma jet (He) | Mouse lung carcinoma and fibroblast cells | Effective induction of apoptosis (but no necrosis) |
| 2011 | **Barezki et al. [42]** | Plasma jet (He) | Human acute lymphoblastic leukemia T-cells (CCRF-CEM and CCL-119) | Prevention of further progression and cell proliferation, dose-dependent cell death. |
| 2011 | **Brulle et al. [43]** | Plasma jet (He, Ne, Ar) | Human pancreatic carcinoma cancer cells (MIA PaCa2-luc) Human Embryonic Kidney cells (HEK293FT) | Inhibition of cancer cells proliferation, synergistic effect via association with radiosensitizer and chemotherapy medication |
| 2012 | **Kaushik et al. [44]** | DBD plasma (Ne) | Human glioblastoma cells (T98G) | Role of plasma exposure time in cell death and micronucleus formation rate, and inhibition of colony formation capacity of cancer cells |
| 2012 | **Iseki et al. [45]** | NEAPP jet (Ar) | Human ovarian cancer cells (SKOV3 and HRA) | Plasma-induced apoptosis and selectivity for cancer cells |
| 2012 | **Partecke et al. [46]** | Plasma jet (kINPen09) (Ar) | Human pancreatic cancer cells (Colo357 and PaTu8988T) Murine pancreatic cancer cells (6606PDA) | Reduction of microscopic residual disease in cancer resections |
| 2012 | **Tuhvatulin et al. [30]** | MicroPlaSter NTP (Ar) | Human colon cancer cells (HCT-116) | p53-dependent apoptosis in cancer cells |
| 2012 | **Vandamme et al. [47]** | FE-DBD (Air) | Human glioblastoma cells (U87MG) Human colon cancer cells (HCT-116) | Highly discrepancy of cell sensitivity between tumor and non-tumor cells and low proliferation rate |
| 2012 | **Yan et al. [48]** | Plasma jet (He) | Human hepatocellular carcinoma cells (HepG2) | Selectivity, inactivation and effective cell death in cancer cells |
| 2013 | **Arndt et al. [49]** | SMD-DBD plasma | Human melanoma cells (Mel Juso, Mel Ei, Mel Ho, Mel Im, Mel Ju, HTZ19) | Dose-dependent cell death |
| 2013 | **Han et al. [50]** | APP jet ($N_2$) | Oral cancer cells (SCC-25) | Time-dependent DSB damage in DNA |
| 2013 | **Köritzer et al. [51]** | SMD-DBD plasma | Human glioblastoma cells (LN18, LN229, U87MG) | Synergistic effects of the combination of plasma and chemotherapeutic agent temozolomide on tumor growth and cell cycle distribution |
| 2013 | **Panngom et al. [52]** | NTP-DBD (Ne) | Human lung cancer cells (H460 and HCC1588) human lung normal cell lines (MRC5 and L132) | High efficiency in lung cancer therapy with mitochondrial dysfunction (morphological changes and reduction in mitochondrial metabolic activity) |
| 2013 | **Utsumi et al. [53]** | NEAPP-activated medium (PAM) (Ar) | Epithelial ovarian cancer cells including: NOS2, NOS3, NOS2TR and NOS3TR (paclitaxel resistant) NOS2CR and NOS3CR (cisplatin resistant) | Enhancing antitumor effect on chemo-resistant cancer cells |

**Table 1.** *Cont.*

| Year of Study | Study Group [Refs] | Plasma Device Type (Injected Gas) | Study Model (Cancer Cell Type) | Key Results |
|---|---|---|---|---|
| 2014 | **Ikeda et al. [54]** | NEAPP jet (He) | Human uterine endometrioid adenocarcinoma cells (HEC-1) Human gastric carcinoma cells (GCIY) | Decreased cell viability of ALDH-high cells in a comparable level to ALDH-low cells |
| 2014 | **Mirpour et al. [55]** | NEAPP jet (He) | Human breast cancer cells (MCF7) Non-tumorigenic epithelial cells (MCF10A) | Enhancing the apoptosis activity in cancer cells after adding oxygen to plasma |
| 2014 | **Plewa et al. [56]** | NTP-DBD (He) | Human colorectal cancer cells (HCT116) | Inhibition of colon carcinoma cell growth in a dose-dependent manner |
| 2014 | **Utsumi et al. [57]** | NEAPP jet (Ar) | Human ovarian cancer cells (TOV21G, ES-2, SKOV3 and NOS2) | Selective cytotoxicity against circulating cancer cells which are resistant to chemotherapy |
| 2015 | **Hirst et al. [58]** | NTP jet (He) | Human prostate cancer cells (BPH-1 and PC-3) | Induction of cytotoxic insult in primary prostate cells leading to rapid necrotic cell death |
| 2015 | **Ikeda et al. [59]** | NEAPP jet (He) | Human endometrioid cancer cells (HEC108 and HEC1) | NEAPP-induced cell apoptosis and more efficient anticancer effects in both ALDH-high and -low cells compared to anticancer drug |
| 2015 | **Ishaq et al. [60,61]** | Plasma jet (He) | Human colorectal cancer cells including: HT29 (TRAIL-resistant cells) and HCT116 | Synergistic effect of the plasma with TRAIL combination treatment in killing drug-resistant cancer cells by inducing apoptosis without toxicity to normal cells |
| 2015 | **Park et al. [62]** | DBD plasma (Ar) | Human breast cancer cells (MCF-7 and MDA-MB-231) Human normal breast cells (MCF-10A and MCF-12A) Human colon cancer cells (HCT-15) Human lung cancer cells (NCI-H1299) | Epigenetic dysregulation of crucial cancer-relevant molecules, including those pertinent to tumor development and apoptosis |
| 2015 | **Lin et al. [63]** | ns-Pulsed DBD plasma | Human nasopharyngeal radioresistant carcinoma cells (CNE1) Human acute monocytic leukemia cells (THP-1) | Enhancing macrophages antitumor effects resulting in stimulation of the immune system |
| 2015 | **Schmidt et al. [64]** | Plasma jet (Ar) | Human melanoma cells (SK-Mel-147) | Increasing anti-metastatic activity in melanoma cells |
| 2015 | **Torii et al. [65]** **Hattori et al. [66]** | NEAPP-activated medium (PAM) | Human gastric cancer cells (NUGC4, SC-2-NU, MKN28 and MKN45) Human fibroblast cells (WI-38) Human pancreatic cancer cells (PANC-1, Capan-2, BxPC-3 and MIA PaCa-2) | Cell apoptosis through ROS generation |
| 2015 | **Weiss et al. [67,68]** | Plasma jet (kINPen09) (Ar) | Prostatic cancer cells (PC-3 and LNCaP) | Significant inhibition of cancer proliferation, as observed for the first time in urogenital cancer |

**Table 1.** *Cont.*

| Year of Study | Study Group [Refs] | Plasma Device Type (Injected Gas) | Study Model (Cancer Cell Type) | Key Results |
|---|---|---|---|---|
| 2016 | **Akhlaghi et al. [69]** | NTP jet (He) | Human lung cancer (LL/2) and normal fibroblast cells (3T3) | Significant reduction of cancer cells viability |
| 2016 | **Kajiyama et al. [70]** | NEAPP-activated medium (PAM) (Ar | Human ovarian cancer cells including: K2 and K2R100 (paclitaxel resistant) and Control cells: TOV21G and ES-2 | Enhancing cancer chemosensitivity |
| 2016 | **Kaushik et al. [71,72]** | Micro-DBD plasma ($N_2$) | Human glioblastoma cells (T98G) Human lung cancer (adenocarcinoma) cells (A549) | Cell mobility promotion in macrophages resulting in stimulation of the immune system |
| 2016 | **Mirpour et al. [73]** | Micro-DBD plasma (He) | Mouse metastatic breast cancer cells (4T1) | Inhibition of the cell migration and cancer metastasis |
| 2016 | **Vermeylen et al. [74]** | PAM and micro plasma jet (He) | Human melanoma cells including: Malme-3M and SK-MEL-28 Human glioblastoma cancer cells including: LN229 andU87 | Variations in sensitivity between different cell lines related to specific mutations; Role of plasma settings and experimental design in the plasma effect |
| 2016 | **Xu et al. [75]** | Plasma jet | Human myeloma cells (RPMI8226 and LP-1 MM) | Induction of myeloma cell apoptosis and enhancing cancer chemo-sensitivity (with bortezomib) |
| 2016 | **Zhu et al. [76]** | Plasma jet | Human breast adenocarcinoma cells (MDA-MB-231) | Synergetic inhibition of cancer cell growth and metastasis due to the combining of drug loaded nanoparticles |
| 2017 | **Binenbaum et al. [77]** | Plasma jet (Ne + Ar) | Murine squamous carcinoma cells (SCC-7) Colon cancer cells (DLD-1) Murine melanoma cells (B-16) | Significant reduction in proliferation of cancer cell lines |
| 2017 | **Chen et al. [78,79]** | Micro-size plasma jet (He) | Human glioblastoma cells (U87MG) | Synergetic treatment effect of short- and long-lived plasma-generated species on cancer cells |
| 2017 | **Li et al. [80]** | DBD plasma (Air) | Human cervical cancer (HeLa) | Induction of apoptosis in HeLa cells via activating ROS generation and mitochondria-mediated apoptotic signaling |
| 2017 | **Yan et al. [81]** | NEAPP-activated medium (PAM) (He) | Human pancreatic adenocarcinoma cells (PA-TU-8988T) Human glioblastoma cells (U87MG) Human breast adenocarcinoma cells (MDA-MB-231) | Significant killing of cancer cells using both plasma-stimulated medium (PSM) and plasma-stimulated buffered solution (PSB) |

**Table 1.** *Cont.*

| Year of Study | Study Group [Refs] | Plasma Device Type (Injected Gas) | Study Model (Cancer Cell Type) | Key Results |
|---|---|---|---|---|
| 2018 | **Lin et al. [82]** | NTP jet (Ar + O$_2$) | Human non-small cell lung cancer cells (A549) Human cervical cancer (HeLa) Human hepatoblastoma (HepG2) Human skin fibroblasts (GM0637) | Synergies of plasma with radiotherapy on cancer cells owing to their combined cytotoxic effects by generating ROS, inducing cell cycle arrest and apoptosis in tumor cells |
| 2018 | **Xu et al. [83]** | NTP jet (He) | Human breast cancer cells (SUM149PT, SUM159PT, MDAMB231, MDAMB436, SKBR3) Human mammary gland epithelial cells (MCF10A) | Deterministic roles on the antitumor efficacy of plasma |
| 2019 | **Azzariti et al. [22]** | DBD plasma (Air + O$_2$) | Human pancreatic ductal cell line (PANC-1) Human sporadic melanoma biopsy specimens Human breast carcinoma cells | Reduction in proliferation and an increase in calreticulin exposure and ATP release, induction of immunogenic cell death via activation of the innate immune system |
| 2019 | **Smolkova et al. [84]** | NTP jet (Air) | Human liver cancer cells (Huh7, Alexander and HepG2) | Induction of apoptotic death in Huh7 and Alexander liver cancer cells and resistance in HepG2 due to the Bcl-2 protein overexpression |
| 2020 | **Adhikari et al. [85]** | Micro-DBD Plasma (Air) | Human melanoma cells (G-361) | Cell apoptosis and autophagy activation due to the decrease in the extracellular pH, leading to a reduction in the intracellular glucose level via inhibition of mTOR and EGF survival pathways |
| 2020 | **Kurita et al. [86]** | NTP jet (He) | Human lung cancer cells (A549) | No induction of strand breaks but induction of 8-oxoG generation in DNA, and no notable reduction in cell viability |
| 2020 | **Pasqual-Melo et al. [87]** | Plasma jet (kINPen09) (Ar) | B16F10 murine melanoma cell | Additive effects of plasma and radiotherapy in cytotoxicity, cell cycle arrest and release of immune-stimulatory products in cancer cells |
| 2020 | **Pranda et al. [88]** | Plasma jet and SMD plasma (Ar) | Human breast adenocarcinoma cells (MDA-MB- 231) Human mammary gland epithelial cells (MCF10A) | Significant role of parameters (type of plasma source and media) in achieving selectivity of cancer cells |
| 2020 | **Zhou et al. [89]** | Two sources: *Invivo*Pen and PAM (He) | Human breast adenocarcinoma cells (MDA-MB- 231) | Similar efficacies in inducing tumor cell apoptosis and suppressing tumor migrative abilities in both sources |

Translation of in vitro results into clinical applications not only needs further investigations of plasma effects on tumorigenic and non-tumorigenic cancer cells, but also necessitates performing in vivo studies in animal models. The early in vivo demonstration of antitumor effect of LTP plasma was performed by Vandamme and co-workers [90,91] with presenting preliminary results on plasma treatment on U87 glioma cancer in both high-radio and chemo-resistant xenograft mouse models. This study highlighted very

promising potential of plasma treatment as an anticancer treatment with little or no toxic side effects. In 2010, Keidar et al. [27] treated bladder and B16/F10 melanoma cancers in subcutaneous xenograft animal models and observed that the tumors with initial size of less than 5 mm disappeared completely; however, larger tumors underwent a reduction in size and maintained their size even after three weeks post treatment. Table 2 listed the main in vivo investigations since the first report in 2010, indicating a fast growth in administration of LTP in cancer treatment. These studies have shown that LTP treatment helps to improve tumor control, stabilize or eradication of tumor volume, as well as improving animal survival. More importantly, plasma treatment was reported in animal models to have the ability of selectively damage on targeted cancer cells without affecting normal surrounding tissues. The reader is referred to [90,92–99] for more details.

**Table 2.** Summary of in vivo preclinical and clinical studies in chronological order, exploiting different types of plasma sources (devices) and study models for cancer treatment, along with the key results obtained from each study.

| Year of Study | Study Group [Refs] | Plasma Device Type (Injected Gas) | Study Model (Cancer Cell Type) | Key Results |
|---|---|---|---|---|
| 2010 | **Vandamme et al. [90]** | μs-pulsed DBD jet | Human glioma (U87-luc) bearing mice | Reduction of tumor volume |
| 2011 | **Keidar et al. [27]** | Plasma jet (He) | B16 and subcutaneous bladder cancer tumors (SCaBER) xenografted in C57Bl6 mice | Reductions in tumor volumes and improving animal survival |
| 2011 | **Kim et al. [92]** | Micro-size plasma jet (He) | B16F0 melanoma tumor in C57BL/6J mouse | Inhibition of tumor growth in four-time treatment plan and no antitumor effect in one-time treatment |
| 2011 | **Vandamme et al. [91]** | NTP-DBD (Air) | Human glioma U87-MG (chemo-resistance) xenografted in mouse | Significant decrease of tumor volume and enhancement of life span |
| 2012 | **Brulle et al. [43]** | Plasma jet (He, Ne, Ar) | Human pancreatic carcinoma (MIA PaCa2-luc) xenografted in mouse | Reducing tumor proliferation and decreasing tumor weight |
| 2012 | **Vandamme et al. [30]** | NTP-DBD (Air) | Human glioma (U87-luc) grafted in mouse | Induction of apoptosis in whole tumor, significant reduction in tumor volume and accumulation of cells in S phase of cell cycle suggesting an arrest of tumor proliferation |
| 2013 | **Daeschlein et al. [93]** | Plasma jet (kINPen09) (Ar) | B16-F10 skin melanoma implantation in C57BL/6N mice | Significant delay in tumor growth |
| 2013 | **Utsumi et al. [53]** | NEAPP-activated medium (PAM) (Ar) | Epithelial ovarian cancer cells (NOS2 & NOS2TR) grafted in mouse | Enhancing cancer chemo-sensitivity |
| 2015 | **Chernets et al. [94]** | ns-pulsed DBD plasma | B16 orthotopic melanoma in C57BL/6 mouse | Tumor eradication |
| 2015 | **Hattori et al. [66]** | NEAPP-activated medium (PAM) | Human pancreatic cancer cells (Capan-2) tumor xenografted in nude mouse (BALB/c) | Significant decrease of pancreatic tumor volume |
| 2015–2016 | **Schuster et al. [95] Metelmann et al. [96]** | kINPen clinical plasma source (He) | 21 patients suffering head and neck cancer | No sign of an enhanced or stimulated tumor growth under influence of plasma treatment |
| 2016 | **Mirpour et al. [73]** | Micro-size plasma jet (He) | 4T1 grafted tumor in BALB/c mouse | Induction of apoptosis in the tumor cells and inhibition its growth |
| 2017 | **Binenbaum et al. [77]** | Plasma jet (Ne + Ar) | Human melanoma tumor in C57/bl mice | Significant reduction in tumor volume |

**Table 2.** *Cont.*

| Year of Study | Study Group [Refs] | Plasma Device Type (Injected Gas) | Study Model (Cancer Cell Type) | Key Results |
|---|---|---|---|---|
| 2018 | **Schuster et al. [97]** | kINPen plasma jet (He) | 20 patients suffering from locally advanced squamous cell carcinoma of the head and neck | Clinical point of view: no risk of severe side effects of applying plasma in cancer patients for palliation |
| 2019 | **Jablonowski et al. [98]** | Two sources: kINPen09 and PS-MWM | Oral Mucosa B6C3F1 mouse | More overt macroscopical and histological lesions, losing more weight in mice, more efficiency of high-temperature PS-MWM than kINPen09 |
| 2020 | **Zhou et al. [89]** | Two sources: In vivo Pen and PAM (He) | Human breast cancer grafted tumor (MDA-MB- 231) in BALB/c mouse | Comparison of two different treatments in preserving mice viability and suppressing tumor growth |

In this article, following the overview of primary in vitro and in vivo studies on antitumor effects of LTP, we briefly outline the plasma sources and devices which have been developed for medical applications, especially those dedicated to cancer treatment. The important mechanisms of interactions between the plasma-generated species with cellular systems and cancer cells will be subsequently described. We will focus on the role of ROS/RNS in cancer inhibition by different mechanisms and pathways involved in those mechanisms that could lead to find novel strategies and anticancer therapies with improved efficacy and safety. Finally, we will discuss new in-sights into the clinical translation of plasma-based cancer treatments and the challenges of transferring plasma into the body especially for tumors located in deep.

## 2. Low Temperature Plasma Sources for Medical Applications

There are a vast number of LTP devices constructed and developed in research laboratories and also commercially available for in vitro and in vivo studies [99,100]. They can be operated using a wide range of tunable process parameters, such as power, voltage, frequency, and injected gas. They can have different structures and configurations and thus different physical discharge concepts and approaches are applied to them, such as dielectric barrier discharge, non-equilibrium atmospheric pressure plasma jet, and spray torch, just to mention a few. The type of plasma devices that was used in the featured studies is indicated in Tables 1 and 2. These devices vary from having a simple construction that only uses one high voltage electrode to sophisticatedly engineered systems that use hybrid or array arrangements.

In spite of the variety of devices used, the basic physics of plasma ignition remains the same across them. The existence of air-free electrons around us can initiate the ionization process if they are influenced by a sufficient electric field that can take place in a confined region between electrodes of a plasma source. Free electrons start to accelerate, and they will then experience a number of collisions with other gas atoms or molecules. Electrons with the energy level necessary for ionization lead to a release of more electrons, and under certain circumstances, the ionization process continues and is sustained, which is known as an electron avalanche. An electron avalanche then induces gas breakdown that causes a series of processes, such as gas molecule dissociation and excitation as well as photon emission. These processes form the previously mentioned plasma species which are chemically reactive. Moreover, because LTP devices operate in an open atmosphere environment, plasma interacts with the air components and forms ROS/RNS even before interactions with cells. Further, plasma also consists of short-lived reactive species that can initiate a cascade of chemical reactions in the cell, which then drives biological outcomes (Figure 1), which will be discussed in next section. As shown schematically in this figure, the contributions from major areas of science and technology are key to the success of

plasma medicine. It is not indicated in the figure, but rapid progress is being made in using mathematical algorithms and computational tools for plasma medicine which is a big step towards achieving an understanding of the clinical implication of LTP devices.

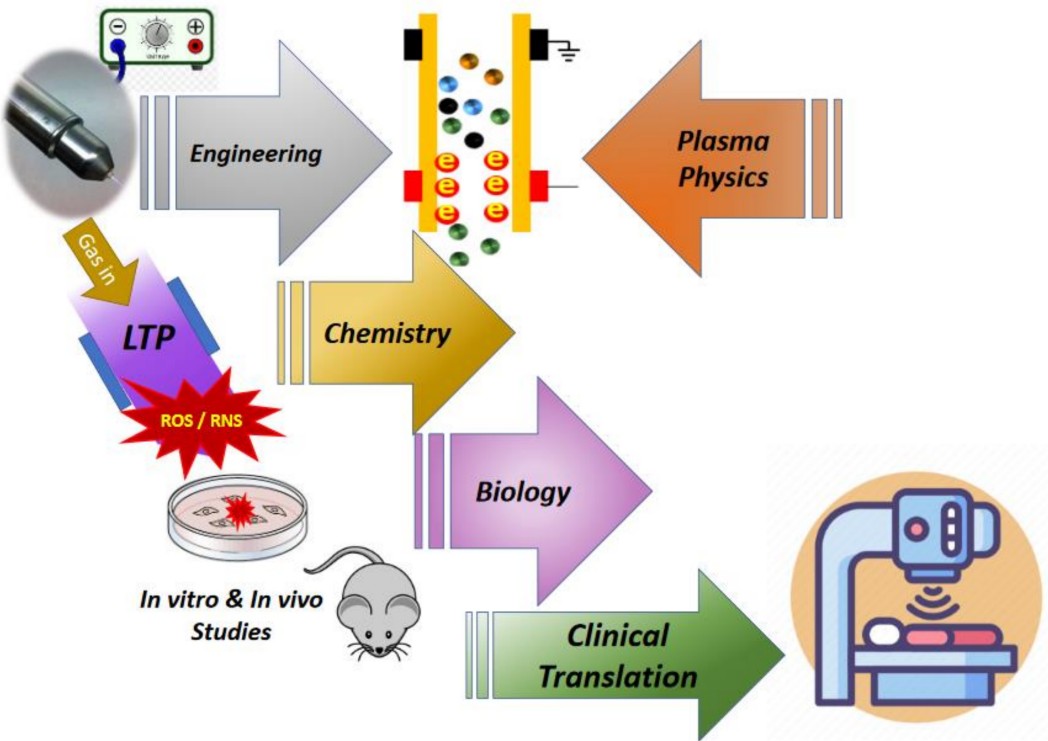

**Figure 1.** Schematic representation of low-temperature plasma (LTP) device and plasma medicine/oncology enabling research intersection between multiple disciplines including engineering, physics, chemistry, biology and medicine.

As mentioned above, many process parameters are involved during the plasma's interaction with biological targets; therefore, choosing the ideal parameter combinations for obtaining desired biological effects is very challenging due to the complexities of cancer cell dynamics. However, the recent introduction of machine learning, which is a branch of artificial intelligence, can dramatically help with providing predictions of plasma treatment-induced changes occurring in cellular systems [101,102]. The optimal plasma parameter choices that could be ideal for specific outcomes of plasma treatment obtained from the predictive modeling could be used to set process parameters of the LTP device. Having the right setting for the device would further strengthen the advantage of using this type of treatment in oncology. A number of in vivo studies has already demonstrated the treatment to be useful for inducing cancer cell damage and avoiding healthy tissue alteration (see Table 2). It took about a quarter of a century for researchers with different expertise from constructing the first lab-based prototypes of an LTP device until the first treatment of human patient in the clinical set-up [103].

For clinical applications, it is important to precisely deliver the desired type and dose of ROS/RNS to the treated target. Therefore, in vitro and in vivo studies are often accompanied by a variety of characterization tools to gain information on the physical (i.e., plasma temperature, power, UV radiation, electromagnetic field) and chemical properties (ROS/RNS and any toxic species) of plasmas [12,104]. A detailed description of these properties is not only important for providing the fundamental mechanisms involved in plasma interactions with the cells, but it is essential to control specific biological responses. The assessment of all risk factors of LTP treatment has to be considered in any pre-clinical trial. Even basic information such as temperature is incredibly important because it is the

essence of these sources for heat-sensitive biological tissues as well as the clinical setup for patient treatment.

These investigations disclosed that beside conventional therapeutic treatments for cancer, LTP may be considered as a promising emerging therapeutic tool for cancer treatment due to its unique biophysical comportment. Prominent advantages of plasma-based cancer treatments compared with conventional therapies are including the ability to originate considerable amounts of reactive species in human cells as the main cause cancer cell death and tumor growth inhibition, high potential of selectivity toward cancer cells, less likely to cause drug-resistance effect [105] and less long-term side effects in cancer patients as used in clinic. Other significant benefit of plasma-based techniques for cancer therapy is their ability to be applied by two different approaches. Generally, in plasma medicine two treatment approaches are introduced using direct and indirect plasma sources [99,106]. In the former, the target is electrically grounded and thus any plasma current induced by charged plasma species can flow through it. This is not the case for indirect plasma sources, which have effects that are influenced by chemical species rather than the physical aspects of the source. It is also important to notice that sometimes direct treatment is defined as a method in which the plasma is in direct contact with the biological target [103]. In that case, all plasma species and their synergistic interactions can take an active role in biological effects. In contrast, in the indirect treatment, only plasma species that still remain after removing ionizing radiation are delivered to the biological target. One such example can be a plasma activated medium (PAM), which is a liquid exposed to plasma before being injected into the biological system [22]. Liquid media such as cell culture and Ringer's lactate solutions, saline and water are the most commonly used media for generating reactive species [107,108]. A study of Oh et al. [109] using tissue and fluid models indicated that direct plasma jet treatment delivered more ROS/RNS and molecular oxygen than the indirect treatment through an agarose medium, which was used as a tissue model, in identical exposure times. The former only delivered ROS/RNS during the plasma jet ignition, while the latter continued to deliver ROS/RNS into a nearby medium long after the plasma was quenched. These results also showed that in the context of direct and indirect plasma treatments of biological fluids and tissues, the type of injected gas (He or Ar) could strongly influence the concentrations of reactive species [109,110]. One of the benefits of PAM is that if it is stored under proper conditions, it will not degrade and lose its chemical properties and then can be applied later for treatment.

Although some primary clinical studies on human patients have been already performed, extensive clinical applications of LTPs still need more detailed investigations on variety of cancer cell lines in both in vitro and in vivo levels. Many parameters, like treatment (or exposure) time, optimal dosage of plasma inside the tissues, tissues thickness, diffusion and penetration depth of reactive species, and cellular damage distribution in biological matters play deterministic roles in the cell-death induction and antitumor efficacy of plasma. Parameters including tissue thickness and tissue properties such as roughness and conductivity, as well as plasma source parameters such as nozzle shapes and the gap distance between the tip of the capillary and the target may also lead to variation of the plasma characteristics such as concentration of reactive species and gas temperature. They can subsequently change the effectiveness, distribution and penetration of the plasma on/in the tissue. Studies on the transportation and distribution of ROS/RNS in an agar tissue phantom after plasma treatment clearly and visually indicated that plasma-generated ROS and RNS were accumulated in the tissue phantom after treatment and then continued to diffuse over and across the tissue [111]. Additionally, clinical application of LTP requires plasma sources to only target and acutely damage cancerous regions of tissue, leaving neighboring normal tissues undamaged. Furthermore, prior to any clinical applications of plasma as a modality for cancer therapy, defining the dose and its quantitative assessment, and finally quantitatively associate dose with the cancer cell killing effect are some of the challenges that require further studies [112].

### 3. Mechanisms of Action of Plasma-Generated Species in Inhibition and Treatment of Cancer

Ionizing radiation, including gamma- and X-ray photons, electrons, alpha particles, and other heavy ions, is one of the most commonly types of radiation applied for cancer treatments. Absorption of high-energy ionizing radiation in cells and tissues induces excitation and ionization of water molecules, which are constituting $70-80\%$ of the cell structure. Thus, majority (over 66%) of the radiation energy deposited in the cell is absorbed initially by water molecules [113]. This is leading to the immediate formation of free radicals such as hydroxyl radicals (OH$^\bullet$), hydrogen (H$^\bullet$), $H_2O_2$ and hydrated electrons, which can react with significant biomolecules like DNA [114]. Hydroxyl free radicals carry no charge, but have a strong affinity for electrons causing to remove hydrogen atoms from biomolecules. For instance, abstraction of deoxyribose hydrogen atoms from DNA initiates at least one pathway, which resulting in the production of a DNA strand scission [115–118]. Free radicals and electrons can also interact with other surrounding molecules like oxygen to generate the highly reactive secondary free radicals and ROS, particularly the superoxide anion radical ($O_2^{\bullet-}$) [119]. Superoxide radical can respectively interact with nitric oxide (NO) to form different RNS like peroxynitrite (ONOO$^-$), which produces cellular dysfunction by S-nitrosylating proteins. Biological mechanisms behind the effectiveness of ROS/RNS in cancer treatment with ionizing radiation have been extensively explored earlier in many studies [29]. High-energy radiation-induced ROS are generated in a very short span of time (shorter than $10^{-8}$ s) [114,120,121] in irradiated cells and indirectly induce damage in mitochondrial proteins, as well as both nuclear DNA (nDNA) and mitochondrial DNA (mtDNA). Ionizing radiation also stimulates an increase in the production of endogenous ROS by mitochondria in the irradiated cells, which potentially leads to mitochondrial dysfunction. The remains of damaged mitochondria could generate or leak more ROS inside the cell, although the damaged mitochondria are removed by mitophagy.

Here, it is important to briefly review the effect of high-energy ionizing radiation on cells, since LTP regulates some similar pathways to cell-killing effect as radiation does. Although besides the ROS produced during water radiolysis and ROS production by mitochondria, there are ROS/RNS generated inside the plasma jet which play multiple roles in signaling cascades and mediates apoptosis [87,119,122]. The OH radical is the major common physicochemical factor which is numerously produced in both plasma and radiation treatments. Under exposure of LTP, generated OH radicals in gaseous form are transferred to the medium. These radicals are the major mediator and responsible for DNA damage in cells [123]. Arjunan et al. [124] has outstandingly reviewed and discussed various plasma-generated ROS/RNS involved in DNA damage, characterized DNA damage and their associated cellular responses induced by atmospheric pressure plasma jet [125]. Interestingly, low levels of ROS/RNS play an important role in vital physiological processes to promote cell survival, proliferation and migration, while excessive ROS levels contribute to accumulating the oxidative stress and finally the initiation and execution of apoptosis [126,127]. Extensive research has shown that these cellular responses can be initiated by severe oxidative DNA damage [128–130]. On the other hand, there is an increasing number of studies verifying the important role of intracellular ROS levels in plasma treatment of cancer cells. Within the cell, plasma-derived ROS can oxidize proteins involved in metabolic pathways, proteasome activity and mitochondrial respiration [131]. In addition, plasma can cause double-strand DNA breaks [50,58] that if irreversible, can lead to cell death [132]. This section mainly focuses on the role of mitochondria in continuous endogenous production of ROS following exposure to radiation or LTP treatment and its relationship to the biological effects.

#### 3.1. Production of Endogenous ROS without Plasma Exposure

Under normal conditions without exposure of cells to high-energy radiation, mitochondria, peroxisomes and endoplasmic reticulum (ER) mostly contribute to the production of

endogenous ROS in cells. Mammalian mitochondria are highly dynamic primary intracellular organelles that have a crucial role in cell metabolism and variety of other additional functions in apoptosis, iron-sulfur (Fe-S) proteins cluster generation and regulating of intramitochondrial calcium concentration [133]. Each mitochondrion contains numerous copies of a circular mitochondrial genome (mtDNA) encoding 13 imported proteins required for electron transport chain (ETC) activity and respiration. All other mitochondrial proteins are nuclear-encoded and are synthesized on cytoplasmic ribosomes [134]. Commonly ETC, tricarboxylic acid (TCA) cycle, oxidative phosphorylation and mitochondria are the cell's principal source of energy. ETC is located on inner membranes of mitochondria and is essential for a number of vital processes including the generation of adenosine triphosphate (ATP). The TCA cycle (also known as the Krebs cycle) is a three-stage process which occurs in the mitochondrial matrix for oxidation of carbohydrates, lipids, and amino acids. This chemical process produces required intermediates NADH (nicotinamide adenine dinucleotide) and $FADH_2$ (flavin adenine dinucleotide) which are electron-rich donors for entering the ETC on the mitochondrial inner membrane for ATP production. Mitochondria possess sensors for molecular oxygen and nutrient levels and contain a number of enzymes like mitochondrial manganese superoxide dismutase (SOD2) to transform oxygen to superoxide or its derivative hydrogen peroxide ($H_2O_2$) radicals. These reactions occur in the ETC when electrons react with $O_2$ as the final electron acceptor resulting in the generation of $O_2^{\bullet-}$ radicals, which is the primary ROS generated in mitochondria (Figure 2). $O_2^{\bullet-}$ radicals are converted by mitochondrial SOD2 into $H_2O_2$, which can be turned into $OH^\bullet$ radical via the Fenton reaction [135]. Electron transfer is linked to the ejection of $H^+$ out of the mitochondrial matrix into the inter-membrane space, creating a proton gradient that drives the production of ATP through oxidative phosphorylation. Thus, consequence of the energy production process is the generation of ROS byproducts. Later, $H_2O_2$ is converted to $H_2O$ and $O_2$ by the action of catalase [136,137].

As the main source of cellular ROS, mitochondria produce up to ninety percent of ROS in normal living cells. Although it rarely happens, $O_2^{\bullet-}$ and $H_2O_2$ may leak into the cell cytoplasm due to the disturbance of mitochondrial homeostasis because of the premature leakage of electrons mainly from defective ETC-related proteins complexes. These leaked ROS can react with important biomolecules, leading to the activation of oxidative stress responses to counteract the ROS. The imbalance of ROS in mitochondria can cause mitochondrial dysfunction which is the decisive factor in the pathways of cell apoptosis. Radiation causes further leakage of electrons from the ETC, excess ROS production and therefore results in further leakage of ROS by mitochondria, which will be more discussed in subsequent section.

Two other intracellular organelles contributing to the generation of ROS in normal cells are peroxisomes and ER. Peroxisomes are dynamic and metabolically active organelles that produce ROS in different metabolic pathways, including β- and α-oxidation of fatty acids, photorespiration, nucleic acid and polyamine catabolism and ureide metabolism. They also contain xanthine oxidase and the inducible form of NO synthase, which produce $O_2^{\bullet-}$ and $NO^\bullet$, respectively. The capacity to rapidly produce and scavenge $H_2O_2$ and $O_2^{\bullet-}$, then increasing the formation of ONOO and $OH^\bullet$ radicals (via Fenton reaction) helps peroxisomes for regulating dynamic changes in ROS levels [138]. ER is a continuous membrane system which constitutes more than half of the membranous content of mammalian cells and plays an important role in calcium storage, lipids metabolism, biosynthesis and transport of proteins and lipids. It is the place for folding and post-translational modifications of newly synthesized proteins. During protein folding process, formation of improperly paired disulfide bonds can lead to the accumulation of misfolded proteins aggregates and stimulation of unfolded protein response to facilitate correct protein folding. ER contains two enzymes, i.e., protein disulfide isomerase (PDI) and ER oxidoreductin 1 (ERO1), which are used for disulfide bond formation in the misfolded proteins, thereby folding them correctly. $H_2O_2$ is generated as a byproduct of oxidative protein folding

process in ER. Glutathione is an essential antioxidant in ER which can reduce the formation of improperly paired disulfide bonds [139].

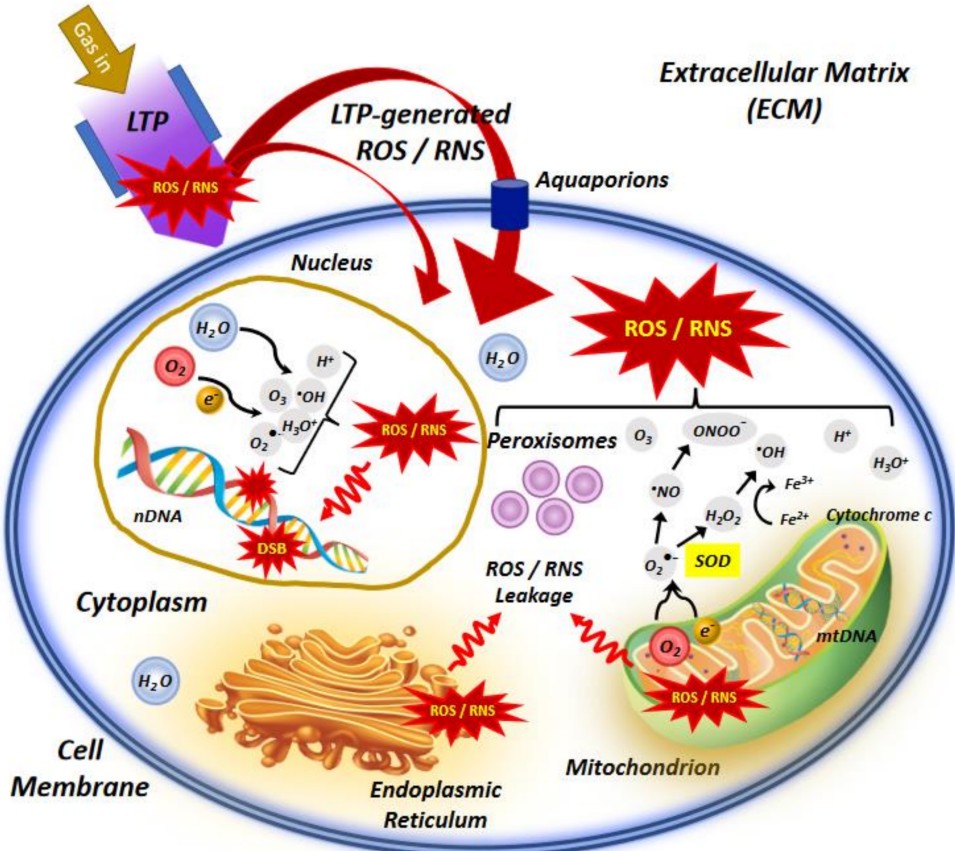

**Figure 2.** An illustrative representation of low temperature plasma (LTP) interaction with the cell, indicating the main molecular mechanisms involved in application of LTP in cancer treatment. In the extracellular matrix (ECM), LTP-generated species can penetrate the membrane of cells and organelles via lipid peroxidation, which leads to pore formation in the cell membrane and facilitates diffusion of reactive species into the cell. This effect may be enhanced in cancer cells due to reduced levels of cholesterol which is responsible for membrane stability and fluidity. Furthermore, extracellular ROS/RNS, specially $H_2O_2$ can pass aquaporins which are often increased in cancer cells and help transition of ROS/RNS via the cell membrane. Inside the cell, major intracellular sources of ROS/RNS are mitochondria, peroxisomes and endoplasmic reticulum (ER). Some species like $O_2^{\bullet-}$ and $H_2O_2$ may leak into the cell cytoplasm due to the disturbance of mitochondrial homeostasis. The imbalance of ROS/RNS in mitochondria can ultimately damages mitochondria causing mitochondrial dysfunction and trigger apoptosis. Increased levels of ROS/RNS by exposure to LTP can also destruct the antioxidant system and limit its protective effect against oxidative stress. Moreover, inside the nucleus, ROS/RNS may attack nearly all significant macromolecules like DNA and induce double strand breaks (DSBs) in nuclear DNA (nDNA).

### 3.2. Production of Mitochondria-Dependent ROS after Plasma Treatment

Depending on the type of human cells, mitochondria occupy a fairly considerable fraction of cell volume (4–25%), which renders them a likely target of radiation traversal through the cell [137]. There are several copies of mtDNA in mitochondria, which code for ribosomal and transfer ribonucleic acid (rRNA and tRNA) and many other essential proteins for sustaining mitochondrial structure and functions [107,140,141]. Other required proteins for mitochondria are encoded by the nDNA. More importantly, as the powerhouse of the cells, mitochondria consume about 90% of the body's oxygen through aerobic res-

piration and are the richest source of ROS. As shown in Figure 2, they divert about 1–5% of electrons from the respiratory chain to the formation of $O_2^{\bullet-}$ radicals by ubiquinone-dependent reduction [142]. Accordingly, exposure to any physical agents or carcinogenetic chemicals, like high-energy radiation in radiation therapy, pharmaceuticals in chemotherapy and LTP in plasma oncology could stimulates the domino effect on the ROS burst in the mitochondria. Various intracellular and extracellular signals induced by LTP-mediated ROS in mitochondria, increase their transmembrane potential and promoting the release of pro-apoptotic factors including cytochrome c. This process is regulated by the B cell lymphoma 2 (Bcl-2) protein family and ultimately leads to the activation of the caspase cascade [37]. Both in vitro and in vivo studies by Arndt et al. have shown that exposure of human melanoma cells to LTP initiated pro-apoptotic changes like Rad17 and tumor suppressor phospho-p53 phosphorylations, cytochrome c release and cleaved-caspase-3 activation, leading to improved wound healing [49,143].

Several observations have suggested that mtDNA could be a key molecule involved in plasma-induced mitochondrial dysfunction. Overproduced ROS accumulated in mitochondria may target the mtDNA polymerase $\gamma$ activity required for replication and repair of mtDNA, thereby resulting in reducing its repair capacity, damage and mutation of mtDNA [144]. They also may modify the assembly of large protein complexes and alter the proper expression of proteins which are required for critical mitochondrial and cellular functions [139]. The accumulation of damaged mtDNAs and mitochondrial proteins inhibits mitochondrial functions, including the maintenance of a stable mitochondrial membrane potential and ATP production. Subsequently, excess ROS accumulated in mitochondria in plasma-irradiated cells cause mitochondrial collapse and irreversible cell apoptosis, since mitochondria act as the major regulator of apoptosis (a type of programmed cell death) [145]. Mitochondrial dysfunction is associated to carcinogenesis, cancer progression and metastasis [146] and mitochondrial pathways such as ROS/RNS signaling or $Ca^{2+}$ homeostasis which significantly contribute to the alteration of energy metabolism in cancer cells [147,148].

These dysfunctions are leading to an increase and continuous leakage of the mitochondrial ROS inside the whole cell and subsequently amplification of damages to nDNA and mitochondria. The presence of mitochondria with damaged mtDNA and oxidized proteins due to radiation-induced ROS production may provoke higher accumulation of oxidative and other types of damages in the cell [140,149,150]. However, dysfunctional mitochondria (those containing damaged mtDNA and oxidized proteins) can be eliminated by mitochondria-specific degradation systems called mitophagy. Mitophagy acts as a mitochondrial quality control measure and prevents excess mitochondrial-dependent ROS accumulation in cells after exposure to IR to repress the effect of leakage of ROS from the damaged mitochondria into the whole cell [151].

Other significant impact of radiation on function of mitochondria may take place during mitochondrial fission and fusion cycles. The mtDNA integrity is maintained during the fission and fusion cycles. Many studies have revealed that ionizing radiation stimulates mitochondrial fission in mammalian cells via an increase in Dynamin-related protein 1 (Drp1) [151,152]. Mitochondrial fission is primarily mediated by Drp1 as a main regulator in the division fission process and essential for the activation of mitophagy. Loss of Drp1 triggers genome instability and initiates DNA damage response by disrupting the mitochondrial fission and distribution [153]. Similar to Drp1, Parkin is another regulator of mitophagy, which its expression increases by radiation, triggering the activation of mitophagy in irradiated cells [154]. Moreover, Parkin-overexpressing cells seem to facilitate the removal of damaged mitochondria and to limit excess mitochondrial ROS production to avoid inducing apoptosis in radiosensitive cells [155].

There are several evidences suggesting that the anticancer effect of plasma radiation is predominantly caused by apoptosis-induction mediated by ROS/RNS primarily act in the extracellular matrix ECM [9]. Triggering apoptosis in plasma-irradiated cancer cells can be assumed from the ROS/RNS generated by LTP and added exogenously, even though

some of these species have a very short life time and diffusion length due to their highly reactive nature and will not be able to reach the ECM, particularly in the bulk of a tumor. Changing the structure and function of proteins at the cell surface or in the ECM has been thoroughly investigated [9,156]. Here, we briefly remark that generated ROS/RNS in plasma can oxidize lipids in the cell membrane, reduce the membrane fluidity and produce pore formation. Thus, due to the permeability of the cell membrane, ROS/RNS penetrate the cell and reach to the intracellular compartment (Figure 2). Thus, cell contents may be released to the ECM, as unregulated digestion of cell components in necrotic cells [157]. These modifications at the cell surface can also activate signaling pathways to alter gene expression, cell growth and maintenance [158].

Furthermore, biological mechanisms behind the high selectivity of LTP to induce apoptosis in cancer cells (as reported in many studies summarized in Table 1) can be elucidated here using plasma-generated ROS/RNS in cancerous and normal cells after LTP treatment [49]. Typically, normal tissues differ from tumor in the proportion of cells in each cell cycle phase at a given time, and in tumor tissues most cells are in the proliferative phase [124]. Plasma-generated ROS/RNS interfere with the signaling pathways used by cancerous cells to inhibit cell proliferation by inducing cell senescence. Thus, the proliferative signal turns into an apoptosis-inducing signal in cancer cells manipulated by LTP, while the signaling pathways manipulated by plasma do not exist in normal cells. Yan et al. demonstrated that LTP increased the percentage of apoptotic tumor cells by blocking the cell cycle at the G2/M checkpoint, and this effect was mediated by reducing intracellular cyclin B1 and cyclin-dependent kinase 1 (Cdc2) and increasing p53 which is resulting in p53-dependant apoptosis [30,48,159]. However, the viability of non-tumor cells can also be altered with longer time of exposure to LTP [160].

Another differing parameter between cancer cells and normal cells that contribute to the high selectivity of LTP for inducing apoptosis in cancer cells is the lower levels of cholesterol in the membrane of cancer cells, which facilitate the diffusion of ROS inside the cells. Additionally, the increased number of aquaporins in the membrane of cancer cells lets easier transport of $H_2O_2$ into the cells [161,162]. When $H_2O_2$ is intact, it may enter the intracellular compartment through aquaporins, where it causes depletion of glutathione (GSH). The depletion of antioxidants like SOD2, GSH and glutathione peroxidases (GPx) via plasma exposure causes membrane attack by the superoxide and $OH^\bullet$ radicals. Formation of $OH^\bullet$ in the vicinity of the cell membrane causes lipid peroxidation of membrane and subsequent cell death by apoptosis. However, the extremely short lifetime and short diffusion length of $OH^\bullet$ prevent harm on neighboring cells [163].

*3.3. Oxidative Stress and Gene Expression*

The level of intracellular ROS/RNS is carefully regulated by endogenous antioxidant enzymes such as SOD2, catalase, GSH and GPx, as well as some low-molecular-weight scavenging enzymes like uric acid, coenzyme Q and lipoic acid [164]. At low concentrations, ROS/RNS play an important role as regulatory mediators in various cellular functions and signaling processes. For instance, ROS are critical components of the antimicrobial repertoire of macrophages for bacterial destruction, or NO is involved in endothelial NO-mediated vasodilatation, which influences the function of circulating cells and underlying smooth muscles in a variety of cardiovascular disorders [165]. Whereas, at moderate or high concentrations, when ROS/RNS levels exceed the capacity of the redox balance control system, a phenomenon appears which is known as oxidative stress, referring to the state that ROS/RNS levels can be cytotoxic and lethal for cells. This suggests that the concentrations of reactive species regulate the shift from their advantageous to detrimental effects, yet the concentrations to which this shift happens and exact mechanisms are unclear [135].

Oxidative stress is involved in the development of various pathological conditions such as psoriasis, cardiovascular disease, neurodegenerative disorders, chronic ulcers, and conclusively tumor promotion and progression in cancer. Compared to normal cells, cancer

cells display weaker antioxidant reactions. This property can facilitate selective attack of cancer cells by extracellular plasma-generated ROS/RNS, resulting in severe oxidative damage and cell death. Additionally, the bursts of ROS/RNS may affect directly or indirectly proteins and genes that participate in oxidative metabolism [166]. Perturbations in oxidative metabolism can lead to chromosomal instability in targeted and non-targeted cells as radiation-induced bystander effects [167]. Numerous in vitro studies (see Table 1) have assessed the effects of LTP treatment on gene expression and epigenetics in several cell lines like melanoma and breast cancers [28,49,62,168,169]. Schmidt et al. observed that oxidative stress and alterations in redox state due to LTP treatment can modify the expression of nearly 3000 genes encoding structural proteins and inflammatory mediators, such as growth factors and cytokines. Moreover, plasma-activated medium treatment on melanoma cells caused changes in cellular morphology and mobility and colony formation, but was less effective on apoptosis and cell cycle progression [64,170,171].

## 4. Challenges and Future Perspectives for Clinical Applications

Undoubtedly, tremendous progress in plasma oncology has been made within a short time of the first lab construction of LTP sources. The first studies focusing on cancer treatment were reported in 2007 and, since then, the number of publications has been exponentially growing [106]. Moreover, the latest approval of clinical trials of plasma-based cancer treatment by federal and governmental organizations in several countries has been a great milestone for this type of treatment as they could lead to alternative modalities to fight against cancer. This progress would not be possible if the idea of plasma medicine, which is clearly multidisciplinary one, was not tackled by researchers with expertise in a variety of fields, including physics, engineering, chemistry, biology, and clinical medicine. Despite this rapid progress and the initial clinical trials, many challenges remain that need to be overcome. Here, we will focus only on aspects that can be explored in basic laboratory settings while aspects related to clinical protocols are not discussed here because they are required to follow the regulations according to the specific country.

In order to improve our understanding of fundamental molecular mechanisms induced by LTP, there is a need for real-time, in situ studies that allow for the characterization of plasma species interactions with cells in both in vivo and in vitro experimental settings. Furthermore, having clearly identified both plasma species in LTP, which is in direct contact with the treated target and plasma species present in the target simultaneously, will advance our understanding of these interactions. Significant efforts have been made to detect either plasma species in the plasma itself or plasma species in the target after plasma treatment. Combining information from both detection approaches will provide a comprehensive picture of biochemical reactions. Novel methods will need to be developed to hybridize optical imaging and spectroscopic tools like in situ UV-Vis spectrophotometers that provide qualitative and quantitative measurements of ROS and RNS in the plasma-cell system [109]. One such hybrid methodology can be using optical emission spectroscopy of plasma with fluorescence microscopy of the cell. This unique assembly could enable us to learn about the spatial localization of plasma reactive species as well as their generation, transport, and propagation in real time during plasma treatment.

In addition to laboratory works aiming to advance the description of the effects and reactions of plasma reactive species on cancer treatment, the data models need to be implemented to deepen our understanding and fill knowledge gaps that are challenging to deliver experimentally. Therefore, predictive modeling of plasma interactions can provide immense opportunities to shed light on this challenge. As previously mentioned, predictive modeling with machine learning is an emerging tool in plasma medicine, and its first applications in this field are receiving significant research attention. The implementation of machine-learning-based models can be used to predict a specific outcome of LTP treatment, which opens the possibilities to find the optimal and desired device settings that would be optimal for cancer treatment.

Although plasma-generated reactive species interactions with the cell have been studied and examined in detail, there are still some unexplored areas that remain to be addressed that would allow the LTP sources to be used in clinic. Being a relatively new concept, LTP devices have not yet been radiation dose calibrated and standardized. Other than a few commercial devices that can be purchased and used for investigations, most laboratories have utilized plasma sources designed and built by researchers; therefore, these devices vary due to different geometries, generated power, and range of parameters that can be tuned. Such diversity of devices hinders the direct comparison of results from one lab to another. For standardization purposes, it would be convenient to introduce a way to measure plasma exposure dose, similar to radiation dose in radiation therapy in which high-energy ionizing radiation is applied. Unfortunately, plasma radiation mainly operates based on ROS/RNS actions with the treated target (not to mention other important plasma components and the synergy among all of them, as well as the target-dependent impact), that characteristics are different than ionizing radiation. Therefore, the commonly used radiation dosimeters and monitoring systems cannot be applied for this purpose. Having dosimeter and real-time monitoring systems of both plasma performance and biological target would allow us to estimate an equal effect on target from exposure of plasma from different LTP sources.

Additional aspects, such as in vivo studies evaluating potential long-term side effects and improving the safety for patients and operators, user-friendly operational systems and software, still need to be considered for LTP devices to be translated to medical instrumentation. Despite all the challenges that still need to be overcome for plasma researchers on the way to clinical applications, LTP-based modality has undoubtedly quickly emerged as one of the newest effective therapeutic strategies for cancer.

**Funding:** This research received no external funding.

**Conflicts of Interest:** The authors declare no conflict of interest.

## Abbreviations

| | |
|---|---|
| $°C$ | degree Celsius |
| Ar | Argon |
| ATP | adenosine triphosphate |
| Bcl-2 | B-cell lymphoma 2 |
| $Ca^{2+}$ | Calcium ion |
| CCRT | concurrent chemoradiation therapy |
| Cdc2 | cyclin-dependent kinase 1 |
| CRT | chemoradiation therapy |
| DBD | dielectric barrier discharge |
| Drp1 | dynamin-related protein 1, also called dynamin-1-like protein |
| DSB | double strand break |
| ER | endoplasmic reticulum |
| ERO1 | endoplasmic reticulum oxidoreductin 1 |
| ETC | electron transport chain |
| FE-DBD | floating electrode dielectric barrier discharge |
| GPx | glutathione peroxidase |
| GSH | glutathione |
| $H_2O_2$ | hydrogen peroxide |
| He | helium |
| LTP | low temperature plasma |
| $\mu s$ | microsecond |
| mtDNA | mitochondrial DNA |
| NADH | nicotinamide adenine dinucleotide |
| nDNA | nuclear DNA |
| NEAPP | non-equilibrium atmospheric pressure plasma |
| Ne | neon |

| ns | nanosecond |
| NO | nitric oxide |
| $NO_2^-$ | nitrite |
| NTP | non-thermal plasma |
| $O_2$ | molecular oxygen |
| $O_2^{\bullet-}$ | superoxide radical |
| OH | hydroxide |
| PAM | plasma-activated medium |
| PDI | protein disulfide isomerase |
| PS-MWM | microwave plasma source |
| RNS | reactive nitrogen species |
| ROS | reactive oxygen species |
| rRNA | ribosomal ribonucleic acid |
| SMD | surface micro discharge |
| SOD2 | superoxide dismutase |
| TCA | tricarboxylic acid |
| tRNA | transfer ribonucleic acid |
| UV | ultraviolet |
| UV-Vis | ultraviolet-visible |
| VUV | vacuum ultraviolet |

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
