# Peer review of "Recent Advances in Plasma-Based Cancer Treatments: Approaching Clinical Translation through an Intracellular View"

_biophysica, doi:10.3390/biophysica1010005_

Round 1
Reviewer 1 Report
The review is an interesting summary of the status quo in applying plasma medicine to cancer therapy. It is quite detailed in discussing the effects of directly or indirectly plasma-produced ROS/RNS(RONS) on the cellular level.
However, there is only very little discussion of other plasma effects, both on cellular or higher levels.
Especially, since this paper is submitted to "Biophysica", some discussion of other physical effects present in the treated system is expected from such a review. E.g., a bit more on the problem of diffusion of RONS in the tissue, direct effects electric fields or even electroporation, the role of UV radiation, differences of using direct of direct vs indirect plasma in the second meaning (line 269) shall be discussed. Also, a more clear connection of in-cell mechanisms to the tissue-level and system-level mechanisms (e.g. cell-to-cell signalling or the immune system response) will be a benefit to the reader. Actually, most of these effects are present already in the referernces and presented in the Tables 1 and 2, so the authors shall not have a problem to summarize it more clearly in the main text.
From my point of view, the abovementioned issues are worthy of at least a paragraph (e.g. as another subsection of section 3) to make the picture of the plasma treatment of cancer more complete. However, I understand that the main focus of the authors are on the mechanism concerning RONS on the cellular level and the discussion of the other effects on a similar level of detail would make the paper truly long. Therefore (in addition to the abovementioned amendments) I suggest to modify the title the paper and reformulate the abstract to more clearly state that this review is focused on the intracellular processes. As far as the title is concerned, I suggest, e.g. to replace the "Approaching Clinical Translation" (that is treated much more briefly) with something like "Intracellular Level View" or such.
While I consider these changes as "major", I think authors are quite capable of handling them for the benefit of the reader.
Some minor issues and suggestions follow:
l.89 entire -> entirely
l.105 speed -> velocity? energy?
l.111/112 Such a low ion temperature is not common, even for a temperature of neutral species it is very low. It is typical only for LTP used in direct plasma medicine. Please, clarify.
l.182+ In Table 2, a number of patients in human studies would be interesting.
l.213-216 The mechanisms of streamers and plasma bullets shall be briefly mentioned since they are much more common in atmospheric pressure LTP sources for plasma medicine than a classical stable electron avalanche.
l.589 The statement "plasma radiation mainly operates based on ROS/RNS actions with the treated target" seems to be a bit too strong (since it is quite target-dependent). Please, clarify.
Reviewer 2 Report
The authors have presented an interesting and detailed review although it should be revised in order to get publication status.
1) The authors overlooked the pioneering work Tuhvatulin A.I. et al., Non-thermal Plasma Causes p53-Dependent Apoptosis in Human Colon Carcinoma Cells // Acta Naturae, 2012, V4, №3, pp 82-87. There, the mechanisms of interaction of LTP with carcinoma cells were first investigated and the role of p53 dependent apoptosis was shown. Therefore, a link to this work should appear both in section 1. Background and Motivation (see line 135 and 155), and in Table 1 as follows:
2012 - Tuhvatulin A.I. et al. - MicroPlaSter plasma flow (Ar) - Human colon cancer cells (HTC116) - LTP Causes p53-Dependent Apoptosis in Human Carcinoma Cells
2) In this review, the authors correctly emphasize the role of mathematical tools for predicting and analyzing ROS and RNS in plasma flows, for their use in medical purposes, and for selecting optimal parameters. It would be advisable to provide a link to the work N. Yu. Babaeva et al., Production of active species in argon microwave plasma torch // J. Phys. D: Appl. Phys., 2018, V.51, pp. 464004. devoted to the computational investigation of active species production, which are important for applications in biomedicine. (see line 120)
I think this review can be interesting for a wide range of readers and can be recommended for publication after necessary corrections as suggested.
Round 2
Reviewer 1 Report
After the changes made by the authors, the paper is now more consistent in the title, content and the overall message. The minor issues seem to be resolved as well.
I therefore recommend the paper for publication and I congratulate the authors to the work done.